# Pediatric Rotavirus Hospitalization Rates in the Military Health System Before and During the COVID-19 Pandemic

**DOI:** 10.3390/vaccines13050492

**Published:** 2025-05-02

**Authors:** Matthew D. Penfold, Sarah Prabhakar, Apryl Susi, Michael Rajnik, Cade M. Nylund, Matthew D. Eberly

**Affiliations:** 1Department of Pediatrics, Uniformed Services University of the Health Sciences, Bethesda, MD 20814, USA; sarah.prabhakar.ctr@usuhs.edu (S.P.); apryl.susi.ctr@usuhs.edu (A.S.); michael.rajnik@usuhs.edu (M.R.);; 2The Henry M. Jackson Foundation for the Advancement of Military Medicine, Inc., Bethesda, MD 20817, USA; 3Pediatric Infectious Diseases, Brooke Army Medical Center, Joint Base San Antonia, Fort Sam Houston, San Antonio, TX 78234, USA

**Keywords:** infectious diseases, rotavirus, vaccines, hospitalizations, COVID-19, pandemic

## Abstract

**Background/Objectives:** Rotavirus gastroenteritis is a vaccine-preventable disease that leads to hospitalization in children less than 5 years of age. Immunizations to prevent rotavirus have greatly altered the epidemiology of significant diarrheal illness. It has been reported that routine immunization rates in children were impacted during the COVID-19 pandemic. Contrary to this fact, rates of many childhood illnesses also decreased. **Methods:** The Military Health System Data Repository (MDR) contains the health records of all military beneficiaries. We queried the MDR before and during the COVID-19 pandemic to assess for alterations in immunization rates and hospitalization rates and to assess for risk factors for significant (hospitalizations) rotavirus disease. **Results:** Our study included a cohort of 1.27 million children under the age of 5 years old. There were 186 unique cases of rotavirus-related hospitalizations over the 5-year study period. During COVID-19 Years 1 and 2, there was a decrease in rotavirus-related hospitalizations compared to the pre-pandemic period. During Year 3, there was a return to the pre-pandemic level of rotavirus hospitalization rates. Patients in the northern United States were less likely to be hospitalized from rotavirus when compared to those in the south. The patients at greatest risk were the youngest beneficiaries. Rotavirus vaccination rates declined in this age group during all three years of the pandemic. **Conclusions:** As the pandemic resulted in less frequent rotavirus immunizations in the Military Health System (MHS), there was not an increase in rotavirus-related hospitalizations above the pre-pandemic baseline.

## 1. Introduction

Rotavirus gastroenteritis, a vaccine-preventable disease, causes significant morbidity and mortality in children less than 5 years of age. Prior to the advent of the rotavirus vaccine in 2006, it was responsible for more than 500,000 worldwide pediatric deaths annually [1]. In 2016, ten years after the first licensed rotavirus vaccine, pediatric rotavirus infections were estimated to cause 128,500 deaths and 1.5 million hospitalizations on an annual basis, with the majority occurring in resource-limited settings such as sub-Saharan Africa [2]. In the United States (US), rotavirus was responsible for 55,000–70,000 hospitalizations annually before the introduction of the vaccine, at a rate of 16 per 10,000 children less than 5 years of age. However, after the vaccine, hospitalization rates were drastically reduced by 80%, with reported rates ranging between 0.96 and 4.0 per 10,000 person-years among US children less than 5 years of age from 2012 and 2015 [3,4].

COVID-19 resulted in many behavioral changes. As public health officials began looking to stop the spread of SARS-CoV-2, social distancing, restrictions to movement, masking, and improved personal hygiene were implemented. These are commonly referred to as non-pharmaceutical interventions (NPI). During this time period, acute gastrointestinal illness (AGI) infections and hospitalizations, including those caused by rotavirus, declined [5]. Studies have suggested that the institution of NPIs for SARS-CoV-2 also reduced fecal-oral transmission of viruses that cause AGIs [6,7,8].

From a public health perspective, a negative consequence of the COVID-19 pandemic was the reduction in rates of pediatric vaccinations globally, in the US civilian population, and within the Military Health System (MHS) [9,10,11]. This gap in routine immunizations, such as rotavirus, left many children vulnerable to serious vaccine-preventable illnesses. Children enrolled in the MHS are a cross-section of the US population that lives around the globe and offers a unique opportunity to study vaccine-preventable diseases. Our hypothesis was that changes in the rotavirus immunization rates would result in an increase in rotavirus hospitalizations as we emerged from 2012 to the COVID-19 pandemic.

## 2. Materials and Methods

A repeated monthly cross-sectional study was performed using population-level outpatient and inpatient data from the MHS Data Repository (MDR) to observe differences in rotavirus vaccination and hospitalization rates and risk factors for hospitalization before and during the COVID-19 pandemic. Data from March 2018 to February 2020 was considered pre-pandemic, while the following three years (through February 2023) were divided into COVID Year 1 (March 2020–February 2021), Year 2 (March 2021–February 2022), and Year 3 (March 2022–February 2023). Children included in the study were between 0 and 59 months of age. Patients’ data, their demographics, and the results were captured and aggregated on a month-by-month basis. If an individual patient’s demographics changed during the study period, they were counted under that new category for the given month. There were no minimum enrollment requirements, and as a database study, the children were free to move, leave, and enter throughout the study period.

We identified those who were hospitalized with rotavirus and those who had received rotavirus vaccines using International Classification of Diseases 10th revision (ICD-10) code A08.0. *Rotaviral enteritis* and Current Procedural Terminology (CPT) codes 90680 and 90681. Unique rotavirus cases were defined as individuals with a hospitalization associated with the rotavirus ICD-10 code. The demographics associated with each unique case are from the time of the earliest encounter for that outcome. We recorded the number of these participants who received rotavirus vaccines and those who were hospitalized for rotavirus during the study period. For the full study population, demographics were recorded at the age of the first encounter in the MHS, whereas demographics for the unique cases were recorded at the time of a rotavirus diagnosis (Table 1). This was necessary as the enrolled population changed on a monthly basis, and thus this approach allowed us to report the number of unique individuals in the study as opposed to using an average or a monthly percentage.

Monthly rates of vaccination and hospitalization were calculated by dividing each monthly total by the number of dependents meeting our study criteria each month. Specifically, vaccination rates were determined based on the total number of rotavirus vaccines administered each month to all children under 60 months of age enrolled in the MHS and did not exclude patients who had already received the rotavirus vaccination series or were outside of the vaccination age eligibility window. COVID-19 period rate ratios (RR) with 95% confidence intervals (CI) were calculated using unadjusted and adjusted Poisson regression controlling for several demographic variables, including sex, parent military rank, age, and region. Age was broken into 6 groups: 0–5 months old, 6–11 months old, 12–23 months old, 24–35 months old, 36–47 months old, and 48–59 months old. Both the hospitalization and vaccination data population enrollment fluctuate each month throughout the study as patients move in and out of the Military Health System. Parent military rank was also used to assess the socioeconomic status of our cohort. Multiple geographic regions—North, South, West, and Overseas—were assessed for significance based on the insurance regions used by TRICARE; the primary medical insurance company of the MHS. The TRICARE South region was used as the geographic control, as gastrointestinal illness rates tend to be less seasonal given the more temperate weather when compared to other regions.

In addition to the primary analysis performed using the COVID-19 time periods, we performed a sub-analysis utilizing the typical rotavirus seasons. These were performed from July 2018 through June 2022. Season 1 (July 2018–June 2019) was used as a pre-pandemic comparator, while the following three rotavirus seasons (Season 2: July 2019–June 2020; Season 3: July 2020–June 2021; and Season 4: July 2021–June 2022) were representative of the expected rotavirus seasons after the start of the pandemic.

We conducted this study with the approval of the Uniformed Services University of the Health Sciences Institutional Review Board (USUHS:2020-065—2 October 2021) in accordance with the Declaration of Helsinki. Additionally, waivers of informed consent and HIPAA were approved as the study dealt with pre-existing de-identified medical records from the MDR.

## 3. Results

### Rotavirus Hospitalizations

Our study included a cohort of 1.27 million MHS beneficiaries under the age of 5 years. Among this group, there were 186 unique cases of rotavirus infections in hospitalized children less than 5 years old when combining pre- and post-pandemic data (Table 1). A total of 100 infections requiring hospitalization occurred in males, and 86 occurred in females. During both Years 1 and 2 of the COVID-19 pandemic, there was a significant decrease in rotavirus-related hospitalizations compared to the pre-pandemic period, with an adjusted RR of 0.18 (95% CI: 0.10, 0.34) in Year 1 and an adjusted RR of 0.41 (95% CI: 0.26, 0.65) in Year 2. During Year 3 after the start of the COVID-19 pandemic, there was a return to the pre-pandemic level of rotavirus hospitalization rates with an adjusted RR of 0.84 (95% CI: 0.59, 1.19). In the sub-analysis using typical rotavirus seasons, there was a statistically significant decrease in rotavirus-related hospitalizations during the post-pandemic rotavirus seasons (Seasons 2, 3, and 4) when compared to the pre-pandemic rotavirus season (Season 1). The adjusted RRs were 0.22 (95% CI: 0.13, 0.36) for Season 2, 0.11 (0.06, 0.22) for Season 3, and 0.71 (0.5, 0.99) for Season 4.

Patients who were enrolled in the TRICARE North region of the United States were statistically less likely to be hospitalized for rotavirus when compared to those enrolled in TRICARE South, with an adjusted RR of 0.56 (95% CI: 0.37, 0.86) (Table 2). Younger age was associated with an increased risk of hospitalization, which was most evident in the youngest age bracket of 0–5 months old and persisted through 35 months of age (Figure 1). Patient sex and parent military rank were not statistically significant. A return to the previously demonstrated seasonality of rotavirus-related hospitalizations with peaks in the late winter was observed (Figure 2).

When compared to pre-pandemic vaccination rates, there was a 12–14% decline in rotavirus vaccine administration in the MHS for COVID-19 Years 1–3 with an RR of 0.86–0.88 (95% CI: 0.86, 0.89). Our study demonstrates that after the pandemic was declared in March 2020, there was a notable decrease in rotavirus vaccines in the cohort, especially during the first 1.5 years, with marginal recovery into the third year of the pandemic (Figure 3).

## 4. Discussion

Despite the decrease in delivery of rotavirus immunizations during Years 1–3 of the COVID-19 pandemic, there was not a rise in serious rotavirus infections leading to hospitalization during Years 1 and 2, but during Year 3, there was a return to pre-pandemic levels. The sub-analysis using typical rotavirus seasons showed a similar decrease in hospitalization rates during Seasons 2 and 3, with a rise in Season 4; however, rates did not return to the pre-pandemic seasonal rotavirus baseline. There were statistically significant decreases in rotavirus vaccination rates in Year 1, 2, and even Year 3 of the COVID-19 pandemic. There was also a decrease in rotavirus hospitalization rates during Years 1 and 2, with a return to the pre-pandemic baseline hospitalization rates in Year 3. The decrease in vaccination rates is consistent with a general decline in US vaccine administration since the start of the pandemic [12]. It is possible that early in the pandemic (Years 1–2), the decrease in vaccination rates was related to a decline in in-person routine well visits, leading to subsequently missed vaccines [13].

Over the course of the pandemic, vaccine hesitancy increased not just for SARS-CoV-2 vaccines, but for all vaccines [14]. This pronounced hesitancy may have also contributed to persistently lower vaccine administration later in the pandemic (Years 2–3), even when most Americans were no longer utilizing strict NPIs. Additionally, in terms of rotavirus vaccination, there is a narrow vaccine administration window, as it must be initiated by 14 weeks and 6 days of life and completed before the infant is older than 8 months, with no available catch-up options after that time. In regard to hospitalization rates, it is possible that the broad application of NPIs led to the significant decrease in hospitalizations in Years 1 and 2. In Year 3, the return to pre-pandemic rotavirus-related hospitalization rates in our cohort is unsurprising given the relaxation of NPIs across the globe, children returning to daycare and school, and declining rates of rotavirus vaccination without catch-up opportunities.

Prior to the pandemic in the US, rotavirus had a typical seasonality, with peaks of hospitalizations and infections occurring during the winter months, which disappeared during the first few years of COVID-19 [15]. A return to the previously demonstrated seasonality of rotavirus-related hospitalizations with peaks in the late winter was observed in this study. Another plausible explanation is the biennial (every 2 years) pattern of rotavirus infections that emerged in the United States after the introduction of the rotavirus vaccine [5]. This phenomenon by itself could explain the re-emergence of rotavirus back to pre-pandemic baseline levels, but further surveillance would be necessary to see if the biennial pattern will return in the post-COVID era.

Consistent with this study’s results, the rates of rotavirus gastroenteritis have staged a slower global return to baseline when compared to other respiratory infections and invasive bacterial infections. Prior to the COVID-19 pandemic, rotavirus was the leading gastrointestinal infection responsible for hospitalization and death in pediatric patients [2]. Since the start of the pandemic in March 2020, many countries, including China, Finland, Brazil, Germany, and the United States, have reported significantly fewer rotavirus infections even after they started to reduce pandemic-related NPIs in the latter stages [5,6,7,16,17]. An MHS-affiliated research team also presented MDR data during a similar pandemic time frame (January 2018 through April 2022) using acute gastroenteritis ICD-10 codes for 0–17-year-old individuals. The data demonstrated a similar rise and fall of gastroenteritis inpatient admissions, with a peak in March 2019 and again in March 2022, with a decline in hospitalization rates from January 2020 until January 2021, which was briefer when compared to the rotavirus data in this study [18]. The decrease in rotavirus infections has occurred in contrast to the return of common respiratory viral infections during the same general time period. Common respiratory viruses were far less common during the height of pandemic NPIs, and there was an associated reduction of many invasive bacterial infections in children that often occur after common respiratory infections [19,20]. Similar to rotavirus infections, invasive bacterial infections from bacteria such as *Streptococcus pneumoniae*, *Haemophilus influenzae*, and *Neisseria meningitidis* have resurfaced in children during the same time as the global relaxation of NPIs [21,22,23,24].

Our study reflects the overall decrease in rotavirus immunization rates across the US. Alarmingly, each year of the pandemic was highlighted by a statistically significant decrease in immunizations relative to the years prior to the onset of COVID-19. Before the pandemic, rotavirus vaccines were estimated to prevent 525,000 hospitalizations around the globe annually. One analysis from 2019 demonstrated that rotavirus vaccines could prevent up to 750,000 hospitalizations annually if further efforts to increase worldwide vaccination rates were undertaken [25]. Specific to our population, children enrolled in the MHS have significantly benefited from rotavirus vaccines, as the hospitalization rate amongst this cohort dropped from 14.37 per 10,000 person-years in 2005–2006 to 5.47 per 10,000 person-years in 2008–2009 [26]. In the US, a study using a theoretical model predicted that incremental improvements in rotavirus vaccine coverage could continue to avert the number of severe cases of rotavirus infections by up to 57% [27]. The US, which recommends rotavirus vaccination as part of its childhood immunization schedule, has historically had lower rotavirus coverage rates compared to other childhood vaccines [28]. For example, in 2016 the rotavirus vaccination rate in the US was 74.1%; however, vaccine administration rates for childhood immunizations given at the same well visits, such as diphtheria, tetanus, and pertussis (DTaP), poliovirus, hepatitis B, *Haemophilus influenzae* type b, and *Streptococcus pneumoniae* vaccines, were higher at 83.4–93.7%, 91.9%, 90.5%, 81.8–92.8%, and 81.8–91.8%, respectively [28]. Despite the comparatively lower rotavirus vaccine administration rate, overall hospitalization rates still remained low, ranging between 0.96 and 4.0 per 10,000 person-years among US children less than 5 years of age from 2012 to 2015 [3]. Addressing and closing the gap in all routine childhood immunizations, including rotavirus, will be an important area of focus for medical providers moving forward in order to help keep hospitalization rates from rising above pre-pandemic levels.

We have speculated on reasons for the return to pre-pandemic hospitalization rates, including reduction in NPIs, return to daycare and school, and the loss of community-wide vaccine-derived immunity. All of these certainly increase the risk of a return to pre-rotavirus vaccine hospitalization rates. Based on our results, the risk of severe infection leading to hospitalization remains elevated through 35 months of age. This is consistent with global reports that demonstrate rotavirus as the leading cause of mortality from diarrhea in children under 5 years of age [29]. The risk of severe infection could potentially be exacerbated by missed rotavirus immunizations, which must be started by 15 weeks of age and completed before 8 months of age. Outside of this time frame, rotavirus is not recommended to be a part of a catch-up immunization schedule. This is a new and looming threat that will require continued research to determine if hospitalization rates increase above pre-pandemic levels or result in a shift in the age of patients with severe disease. However, if missed vaccinations create a cohort of older children susceptible to rotavirus disease, this may be ameliorated as older children are less susceptible than young infants to severe disease requiring hospitalization.

Despite the proposed reasons for an increase in rotavirus-related hospitalizations, our study did not see the increase until year 3, and it did not rise above the pre-pandemic levels. A potential explanation as to why the increase was not worse is herd immunity. One study in the US, which evaluated over 9,000 rotavirus gastroenteritis infections, found a substantial indirect vaccine efficacy in unimmunized persons ranging from 35% in 45–64-year-old individuals and up to 79% in children under 1 year of age [30]. Another global study looking at 112 countries found more modest indirect impacts of rotavirus immunization, with vaccine efficacy no higher than 26.7% in the unimmunized [31]. With these studies in mind, infants who were immunized potentially provided adequate protection to their close contacts who were not vaccinated and therefore did not increase hospitalization rates above the baseline in our cohort. It is also reasonable to consider that at the beginning of the pandemic, patients were more conscious of contagious diseases and regularly practiced appropriate NPIs, but those behaviors waned over time [32].

It must be highlighted that the beneficiaries included in the study are from a global population. Thus, NPIs that impacted this cohort were asynchronous and variable. Many factors impacted the modification of NPIs. These include mandates to return to work and school, the acquisition of community-wide immunity through either immunization or infection, and societal factors that limited the continued acceptance of these practices. For example, Year 1 of the pandemic led to significant lockdowns and closures within the US. The second year featured more in-person work and school as the population began to become immunized. In the third year to present, lifestyles returned to pre-pandemic forms. Thus, there was an expected resumption of transmission of pathogens by the fecal-oral route, leading to a return of rotavirus.

The primary strength of this study is the large cohort of over 1 million pediatric patients enrolled in the MHS, including their ICD-10 codes obtained from civilian institutions and military treatment facilities. The MHS pediatric population predominantly lives in the continental United States, but a small proportion of the population lives globally. Interestingly, there was a statistically significant reduction in hospitalization rates in patients enrolled in TRICARE North when compared to TRICARE South. While this study is observational and not designed to address the cause of regional differences, we postulate that the regional variations regarding social distancing and non-pharmacologic interventions such as masking, handwashing, and social distancing likely played a role. Certainly, climate can also impact regional variations in rotavirus rates, as warmer climates are known to have a more constant circulation of rotavirus. Limitations of this study include a reliance on ICD-10 codes, unknown changes in rotavirus testing practices before and after the pandemic, the inability to directly correlate rotavirus testing and ICD-10 codes for the entire cohort, and unknown immunization status of hospitalized patients. Additionally, despite our large cohort, there were relatively few cases to analyze, which may limit the study’s conclusions.

## 5. Conclusions

Despite the reduction in routine rotavirus immunization administration to children in the MHS after the COVID-19 pandemic started, there was no increase in rotavirus-related hospitalizations above the pre-pandemic baseline. In the first two years following the start of the pandemic, there was a statistically significant decrease in hospitalizations related to rotavirus in our cohort, followed by a return to baseline in the third year. These shifts were temporally associated with the initiation of public health strategies meant to reduce the spread of SARS-CoV-2. Social distancing, to include daycare closures, increased hygiene measures, and catch-up immunizations, may explain this finding. Lastly, missed vaccinations during the narrow rotavirus vaccination window may leave many children at risk for severe rotavirus infections during early childhood. Appropriate and timely childhood immunizations are paramount to protecting young children from rotavirus. Continued surveillance is warranted to determine if hospitalization rates return to pre-pandemic levels or result in a shift in the age of patients with severe disease.

## Figures and Tables

**Figure 1 vaccines-13-00492-f001:**
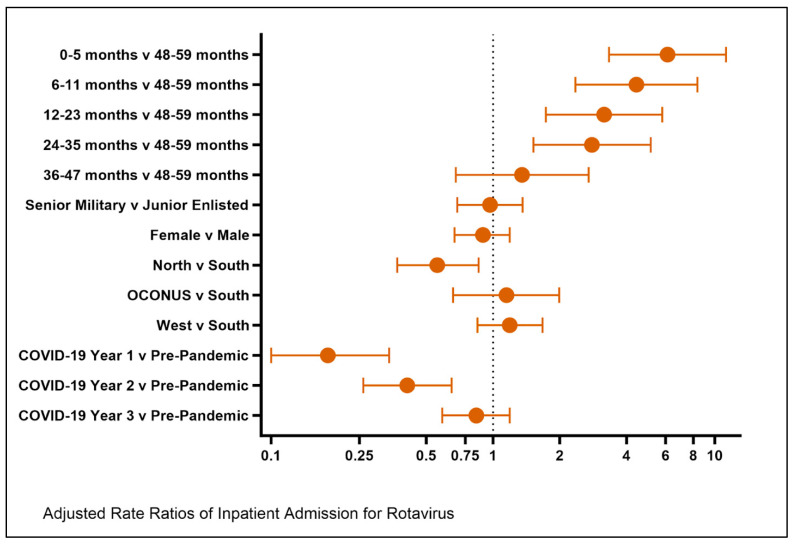
Rotavirus hospitalization forest plot of adjusted rate ratios for inpatient admissions.

**Figure 2 vaccines-13-00492-f002:**
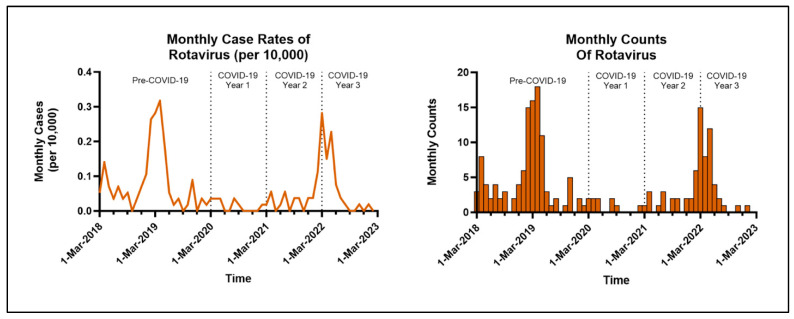
Monthly rates and counts of inpatient rotavirus infections during the study period.

**Figure 3 vaccines-13-00492-f003:**
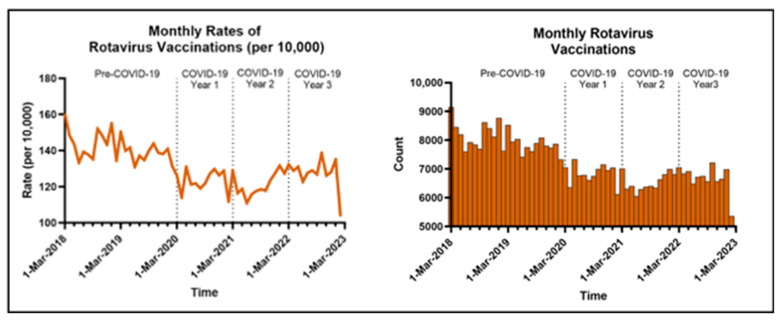
Monthly rates and counts of rotavirus vaccinations during the study period.

**Table 1 vaccines-13-00492-t001:** Demographics at time of first encounter.

		Unique Cases of Rotavirus (n = 186) *	Full Study Population (n = 1,268,604) **
**Age (months old)**	0–5 months	42 (22.6%)	604,871 (47.7%)
	6–11 months	31 (16.7%)	80,395 (6.3%)
	12–23 months	42 (22.6%)	152,975 (12.1%)
	24–35 months	39 (21.0%)	148,146 (11.7%)
	36–47 months	18 (9.7%)	143,572 (11.3%)
	48–59 months	14 (7.5%)	138,645 (10.9%)
**Sex**	Male	100 (53.8%)	648,834 (51.2%)
	Female	86 (46.2%)	619,770 (48.9%)
**Sponsor Rank**	Junior Enlisted	44 (23.7%)	369,208 (29.1%)
	Other	142 (76.3%)	899,396 (70.9%)
**Region**	North	35 (18.8%)	392,933 (31.0%)
	South	60 (32.3%)	376,091 (29.7%)
	West	75 (40.3%)	409,512 (32.3%)
	Outside the US	16 (8.6%)	90,068 (7.10%)

* Age at first rotavirus hospitalization. ** Age at entry into the MHS.

**Table 2 vaccines-13-00492-t002:** Rotavirus Inpatient Results—Adjusted and Unadjusted Risk Ratios (RR).

Rotavirus Hospitalizations			
		Unadjusted RR (95% CI)	Adjusted RR (95% CI)
**Age (reference = 48–59 months)**	0–5 months	6.27 (3.42, 11.47)	6.12 (3.33, 11.23)
	6–11 months	4.51 (2.40, 8.49)	4.43 (2.35, 8.34)
	12–23 months	3.20 (1.76, 5.85)	3.17 (1.73, 5.79)
	24–35 months	2.82 (1.53, 5.19)	2.79 (1.52, 5.14)
	36–47 months	1.36 (0.68, 2.72)	1.35 (0.68, 2.70)
**Rank (reference = Junior Enlisted)**	Senior Military	0.82 (0.59, 1.14)	0.97 (0.69, 1.36)
**Sex (reference = Male)**	Female	0.90 (0.67, 1.19)	0.90 (0.67, 1.19)
**Region (reference = South)**	North	0.57 (0.37, 0.86)	0.56 (0.37, 0.86)
	Outside the US	1.19 (0.69, 2.07)	1.15 (0.66, 1.99)
	West	1.23 (0.88, 1.72)	1.19 (0.85, 1.67)
**Time Period (reference = Pre-COVID-19)**	COVID-19 Year 1	0.18 (0.09, 0.35)	0.18 (0.10, 0.34)
	COVID-19 Year 2	0.41 (0.26, 0.65)	0.41 (0.26, 0.65)
	COVID-19 Year 3	0.84 (0.59, 1.19)	0.84 (0.59, 1.19)

## Data Availability

This data are not open-sourced. As the data are derived from the MHS Data Repository, sharing of data would require a specific data-sharing agreement that would need to be approved by the Department of Defense stakeholders, to include the Defense Health Agency.

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
