# Peer review of "Pediatric Rotavirus Hospitalization Rates in the Military Health System Before and During the COVID-19 Pandemic"

_vaccines, 2025, doi:10.3390/vaccines13050492_

Round 1
Reviewer 1 Report
Comments and Suggestions for Authors
This is a well written article that makes use of a unique dataset to provide information on rotavirus hospitalizations and vaccination before and during the COVID-19 pandemic. It is a worthwhile analysis that will be of interest to readers. That said, there are significant limitations to the methods description and implementation that must be addressed before it should be considered for publication.
- This analysis is undertaken as a cohort study, however the cohort is not adequately described in the methods. What happens to children who were not followed for the full time period? How was the population calculated in situations where children moved between regions mid-year?
- The description and interpretation of rates of vaccination is wholly inadequate. What the denominator for monthly vaccination rates? It would be useful to also present the birth cohort rotavirus vaccination coverage at 12-23 months of age to allow for comparison with other data sources such as NIS. Finally, it would be interesting to look at timeliness of vaccination in this population as the upper age restrictions for rotavirus vaccine have been hypothesized to impact coverage, especially when compared with other routine infant vaccines.
- You used ICD 10 codes to determine rotavirus in the absence of testing, the sensitivity and specificity of which are not great (Pindyck T, Hall AJ, Tate JE, Cardemil CV, Kambhampati AK, Wikswo ME, Payne DC, Grytdal S, Boom JA, Englund JA, Klein EJ. Validation of acute gastroenteritis-related international classification of diseases, clinical modification codes in pediatric and adult US populations. Clinical Infectious Diseases. 2020 May 23;70(11):2423-7.) While you do address this in the limitations, the analysis would be stronger if rates of acute gastroenteritis of any cause were also included. There are several models in the literature of how this can be done (Baker JM, Dahl RM, Cubilo J, Parashar UD, Lopman BA. Effects of the rotavirus vaccine program across age groups in the United States: analysis of national claims data, 2001–2016. BMC infectious diseases. 2019 Dec;19:1-1.)
- "Unique cases" is not defined in the methods.
Author Response
Comment 1: "This is a well written article that makes use of a unique dataset to provide information on rotavirus hospitalizations and vaccination before and during the COVID-19 pandemic. It is a worthwhile analysis that will be of interest to readers. That said, there are significant limitations to the methods description and implementation that must be addressed before it should be considered for publication."
Response 1: Thank you for this very positive feedback! Our team worked together to create this study, write the initial manuscript, and now improve our manuscript so that it meets the desired level of academic rigor worthy of publication. We appreciate your help in making this happen.
Comment 2: "This analysis is undertaken as a cohort study, however the cohort is not adequately described in the methods. What happens to children who were not followed for the full time period? How was the population calculated in situations where children moved between regions mid-year?"
Response 2: Thank you for this prudent point. Our methods section could be improved with further granularity about our population and the data we analyzed.
This is a repeated monthly cross-sectional study meant to simulate a cohort study, but at a population level (as opposed to individual level). The children were included on a month-by-month basis. If any demographics changed (age, region), they would then be counted under that new category for the given month. The demographics were attached to the outcomes at the month level too. There were no minimum enrollment requirements and as a database study, the children were free to move, leave, and enter throughout the study period.
We have updated the manuscript to better describe our cohort in the first paragraph of the methods section.
Comment 3: "The description and interpretation of rates of vaccination is wholly inadequate. What the denominator for monthly vaccination rates? It would be useful to also present the birth cohort rotavirus vaccination coverage at 12-23 months of age to allow for comparison with other data sources such as NIS. Finally, it would be interesting to look at timeliness of vaccination in this population as the upper age restrictions for rotavirus vaccine have been hypothesized to impact coverage, especially when compared with other routine infant vaccines."
Response 3: We appreciate this constructive criticism and have worked to improve our discussion of the vaccination data.
The denominator is the number of individuals enrolled in a given month, which fluctuates each month throughout the study. The numerator each month was purely the number of rotavirus vaccinations that were administered to children <60 months of age and did not account for previous vaccinations, or rotavirus vaccination eligibility based on age.
We don't technically have a birth cohort as part of our study. While we agree that looking at a birth cohort would be interesting. Unfortunately, it would require a different study outside of what our team has accomplished as part of this manuscript.
Finally, as our study was a population level study that aggregated monthly vaccination and hospitalization rates, we did not track individual vaccinations nor their individual histories or the timing of their vaccines.
We clarified the population level of the study, clarified the vaccination counts, and specifically detailed that the denominator changed from month to month.
Comment 4: "You used ICD 10 codes to determine rotavirus in the absence of testing, the sensitivity and specificity of which are not great (Pindyck T, Hall AJ, Tate JE, Cardemil CV, Kambhampati AK, Wikswo ME, Payne DC, Grytdal S, Boom JA, Englund JA, Klein EJ. Validation of acute gastroenteritis-related international classification of diseases, clinical modification codes in pediatric and adult US populations. Clinical Infectious Diseases. 2020 May 23;70(11):2423-7.) While you do address this in the limitations, the analysis would be stronger if rates of acute gastroenteritis of any cause were also included. There are several models in the literature of how this can be done (Baker JM, Dahl RM, Cubilo J, Parashar UD, Lopman BA. Effects of the rotavirus vaccine program across age groups in the United States: analysis of national claims data, 2001–2016. BMC infectious diseases. 2019 Dec;19:1-1.)"
Response 4: This is a great point and similar data from the MHS has been previously reported. We updated the our discussion section to include a comparison of the data in an abstract and poster that was presented in 2023 at the Academy Health Annual Research Meeting in Seattle, Washington, USA. This is listed as citation #18 in the updated manuscript. Ultimately, the acute gastroenteritis data followed a similar trend with peaks and valleys reflective of the rotavirus data.
Comment 5: "'Unique cases' is not defined in the methods."
Response 5: Thank you for pointing this out and marks an important definition that we need to include. The manuscript has been updated and “Unique cases” is now defined in the second paragraph of the methods section.
Reviewer 2 Report
Comments and Suggestions for Authors
This manuscript presents an interesting analysis rates of rotavirus hospitalizations in military dependents <5 years of age before and during the COVID-19 pandemic and found that rotavirus hospitalization rates did not return to pre-pandemic levels until the third year after the start of the pandemic. The authors also looked at rates of rotavirus vaccination during the same periods but these data were harder to follow and interpret. Some specific comments are provided below.
Major Comments:
Please provide more information how vaccination status was determined. For children enrolled since birth, the CPT codes for rotavirus vaccination would have been captured by the MHS. However, for the 52.3% of children who entered the MHS at age of 6 months or older, how was the vaccination status determined? Is it possible to restrict the analysis to children who have been enrolled continuously since birth to ensure that all doses of rotavirus vaccine are captured?
What are the numerator and denominator for the monthly rotavirus vaccination rates per 10,000? Is this all children <5 years of age, children age-eligible to have received at least one of rotavirus vaccine, children >8 months of age (e.g. age-eligible to be fully vaccinated)? To fully see the impact of the COVID-19 pandemic on rotavirus vaccination rates, it would be helpful to examine vaccination rates by birth cohort. Children who were born in January and February 2020 (and therefore reached 14 weeks 6 days during the early COVID-19 pandemic) would be expected to be most impacted by the pandemic.
Are the monthly rates of rotavirus vaccination calculated for children who received at least one dose or children who are fully vaccinated (defined based on which rotavirus vaccine received)?
The pre- and post-pandemic periods are defined using March 2020 as an anchor. This time point certainly makes sense especially when looking at vaccine coverage as there is no seasonality. However, when describing rotavirus hospitalization rates, it potentially splits rotavirus seasons into two different years (COVID-19 years 2 and 3 split the peak of the 2022 rotavirus season into separate years). In addition to the COVID-19-defined periods, consider also a secondary analysis comparing rotavirus seasons defined for July-June so that findings can be compared to previously published literature based on rotavirus seasons.
Minor Comments:
Abstract, Conclusions, page 1, line 28: The conclusions state that there was a decrease in rotavirus immunization rates. However, no data on immunization rates in the abstract. If this is a key finding, include immunization rates in the results section of the abstract.
Discussion, page 5, lines 142-144: Also, note that in addition to all doses being administered by 8 months of age, the rotavirus series must be initiated by 14 weeks 6 days.
Author Response
Comment 1: "This manuscript presents an interesting analysis rates of rotavirus hospitalizations in military dependents <5 years of age before and during the COVID-19 pandemic and found that rotavirus hospitalization rates did not return to pre-pandemic levels until the third year after the start of the pandemic. The authors also looked at rates of rotavirus vaccination during the same periods but these data were harder to follow and interpret. Some specific comments are provided below."
Response 1: Thank you for taking the time to review and improve our manuscript. We appreciate your efforts in helping us ensure our manuscript clearly presents our findings to its future readers.
Comment 2: "Please provide more information how vaccination status was determined. For children enrolled since birth, the CPT codes for rotavirus vaccination would have been captured by the MHS. However, for the 52.3% of children who entered the MHS at age of 6 months or older, how was the vaccination status determined? Is it possible to restrict the analysis to children who have been enrolled continuously since birth to ensure that all doses of rotavirus vaccine are captured?"
Response 2: Thank you for asking us to clarify this point as understanding how our vaccination data was derived is incredibly important to the study.
Vaccination status was never determined on an individual level, but rather was a population look at the number of times the CPT codes for rotavirus vaccination were used in children under 5 years of age who were enrolled in the MHS during each month of the study. Since the study was done with aggregated and population level data, using our current methods, we are unable to follow a birth cohort of patients, but we agree that this would be a useful way to look at the data.
The specifics of how the vaccination data was obtained has been clarified in the Methods section.
Comment 3: "What are the numerator and denominator for the monthly rotavirus vaccination rates per 10,000? Is this all children <5 years of age, children age-eligible to have received at least one of rotavirus vaccine, children >8 months of age (e.g. age-eligible to be fully vaccinated)? To fully see the impact of the COVID-19 pandemic on rotavirus vaccination rates, it would be helpful to examine vaccination rates by birth cohort. Children who were born in January and February 2020 (and therefore reached 14 weeks 6 days during the early COVID-19 pandemic) would be expected to be most impacted by the pandemic."
Response 3: We absolutely appreciate what the reviewer is asking as the counts and rates of the vaccination data is important to the context and interpretation of our study.
The numerator was every time a rotavirus vaccine CPT code was captured in a given month. The denominator was the number of children <60 months of age currently enrolled in the MHS. It did not take into account eligibility for the vaccine based on age or prior receipt of the rotavirus vaccination.
While we very much appreciate the reviewer’s point that looking at a birth cohort would be interesting and useful; unfortunately, it would require a study design that is different from what our team has accomplished as part of this manuscript.
Comment 4: "Are the monthly rates of rotavirus vaccination calculated for children who received at least one dose or children who are fully vaccinated (defined based on which rotavirus vaccine received)?"
Response 4: Thank you for asking this important question, which was not clearly outlined in our methods section. The monthly rates were calculated based on the number of rotavirus vaccination CPT codes captured (numerator) over the total number of children <60 months of age enrolled in the MHS during that month (denominator).
We have updated the methods section to ensure our future readers understand how these numbers were calculated.
Comment 5: "The pre- and post-pandemic periods are defined using March 2020 as an anchor. This time point certainly makes sense especially when looking at vaccine coverage as there is no seasonality. However, when describing rotavirus hospitalization rates, it potentially splits rotavirus seasons into two different years (COVID-19 years 2 and 3 split the peak of the 2022 rotavirus season into separate years). In addition to the COVID-19-defined periods, consider also a secondary analysis comparing rotavirus seasons defined for July-June so that findings can be compared to previously published literature based on rotavirus seasons."
Response 5: Thanks for this important feedback! We want to make sure that our readers have a clear understanding of the pandemic timeline, and agree that the rotavirus seasonality is different than the COVID-19 time periods we utilized. In order to address this, we completed a sub-analysis using the July-June rotavirus season that would have overlapped with our COVID-19 time periods. We have included these updates in our Methods, Results, and Discussion. Ultimately, the rotavirus seasons also saw a significant decline in hospitalization rates, similar to our COVID-19 time periods, although the final rotavirus season did not return to the pre-pandemic rotavirus baseline (adjusted RR 0.71; 95% CI: 0.50, 0.99). Since the data was similar to the results using the pandemic years, we opted to discuss it in the text of the manuscript without adding it to our tables or figures.
Comment 6: "Abstract, Conclusions, page 1, line 28: The conclusions state that there was a decrease in rotavirus immunization rates. However, no data on immunization rates in the abstract. If this is a key finding, include immunization rates in the results section of the abstract."
Response 6: Thanks for catching this oversight as the immunization rates are an important aspect of our study. This has been updated in the results section of the abstract.
Comment 7: "Discussion, page 5, lines 142-144: Also, note that in addition to all doses being administered by 8 months of age, the rotavirus series must be initiated by 14 weeks 6 days."
Response 7: The reviewer points out a very important point that the rotavirus vaccine series is time bound both by the age at which it is initiated and that it is completed. This has been updated in the discussion section.